# Disentangling bias between $G_q$, GRK2, and arrestin3 recruitment to the $M_3$ muscarinic acetylcholine receptor

Anja Flöser[1,2†], Katharina Becker[2†], Evi Kostenis[3], Gabriele König[3], Cornelius Krasel[2], Peter Kolb[1]*, Moritz Bünemann[2]*

[1]Department of Pharmaceutical Chemistry, Philipps-University Marburg, Marburg, Germany; [2]Department of Pharmacology and Clinical Pharmacy, Faculty of Pharmacy, Philipps-University Marburg, Marburg, Germany; [3]Molecular, Cellular and Pharmacobiology Section, Institute for Pharmaceutical Biology, University of Bonn, Bonn, Germany

*For correspondence:
peter.kolb@uni-marburg.de (PK);
moritz.buenemann@staff.uni-marburg.de (MB)

†These authors contributed equally to this work

Competing interest: The authors declare that no competing interests exist.

**Abstract** G protein-coupled receptors (GPCRs) transmit extracellular signals to the inside by activation of intracellular effector proteins. Different agonists can promote differential receptor-induced signaling responses – termed bias – potentially by eliciting different levels of recruitment of effector proteins. As activation and recruitment of effector proteins might influence each other, thorough analysis of bias is difficult. Here, we compared the efficacy of seven agonists to induce G protein, G protein-coupled receptor kinase 2 (GRK2), as well as arrestin3 binding to the muscarinic acetylcholine receptor $M_3$ by utilizing FRET-based assays. In order to avoid interference between these interactions, we studied GRK2 binding in the presence of inhibitors of $G_i$ and $G_q$ proteins and analyzed arrestin3 binding to prestimulated $M_3$ receptors to avoid differences in receptor phosphorylation influencing arrestin recruitment. We measured substantial differences in the agonist efficacies to induce $M_3R$-arrestin3 versus $M_3R$-GRK2 interaction. However, the rank order of the agonists for G protein- and GRK2-$M_3R$ interaction was the same, suggesting that G protein and GRK2 binding to $M_3R$ requires similar receptor conformations, whereas requirements for arrestin3 binding to $M_3R$ are distinct.

## Editor's evaluation

This paper investigates the molecular mechanism of ligand bias in G protein-coupled receptors, specifically the $M_3$ muscarinic receptor, which is the property that different receptor agonists favor activation of G protein signaling vs. arrestin-mediated signaling. The interaction with arrestin is promoted by receptor phosphorylation by G protein-coupled receptor kinases (GRK), and GRK recruitment is expected to influence arrestin-recruiting activity of a particular ligand. However, the possibility of a distinct agonist-dependent receptor conformation on GRK association has not been investigated. This study demonstrates that the G protein $G_q$ and GRK2 appear to interact with a similar $M_3$ receptor conformation, whereas arrestin3 interacts with a distinct conformation. This represents a significant advance in understanding the mechanism of ligand bias in G protein-coupled receptors.

## Introduction

G protein-coupled receptors (GPCRs) are membrane-spanning proteins that convert extracellular to intracellular signals. GPCRs can sense a wide variety of different agents, from single photons

in the case of rhodopsin to small proteins that activate chemokine receptors (*Shichida and Imai, 1998*; *Griffith et al., 2014*). The canonical sequence of events during receptor activation (not taking into account precoupling of G protein) is the following: (1) an agonist, increasing the population of active receptor conformations, binds to the orthosteric pocket; this leads to a further conformational change of the transmembrane helix bundle that results in an opening of the intracellular effector-binding cavity; (2) G proteins bind to the intracellular pocket of the receptors, are thereby activated and enabled to transmit and modulate a multitude of signals; (3) G protein-coupled receptor kinases (GRKs) are recruited to the activated receptors, leading to receptor phosphorylation; and (4) arrestins bind to the active conformation of these receptors after being activated by an initial binding to phosphorylated residues at the intracellular end of the receptors. Subsequently, arrestins induce internalization and desensitization of the receptors (*Ritter and Hall, 2009*; *Martí-Solano et al., 2016*; *Scheerer and Sommer, 2017*; *Kenakin, 2019*). In the sequence of events described above, the three GPCR-binding proteins investigated in this work (G proteins, GRKs, and arrestins) recognize agonist-induced receptor conformations independently of each other, as shown in *Figure 1—figure supplement 1*. There is strong evidence that different agonists can stabilize distinct receptor conformations and thereby influence the recruitment of effector proteins leading to 'functional selectivity' or, more specifically, 'ligand bias' to better reflect the nomenclature in the upcoming BJP guidelines (*Costa-Neto et al., 2016*; *Martí-Solano et al., 2016*; *Wingler et al., 2019*; *Xu et al., 2019*; *Kenakin, 2019*). It has been shown before that receptor conformations exist that are preferentially recognized by either arrestins or G proteins. What these different receptor conformations might look like is a hot topic of current research (*Wingler et al., 2020*; *Suomivuori et al., 2020*). The recent study of *Stoeber et al., 2020* has shown that agonist-dependent recruitment of GRK2 to an opioid receptor exists. This leads to the question whether a functional bias at the level of arrestin recruitment (*Carr et al., 2016*) is affected by both receptor recruitment rates of GRKs and arrestins. Experimentally it has been difficult to distinguish between both events. Here, we set out to address agonist selectivity independently for all three receptor interaction events by comparing agonist-induced binding of $G_q$ proteins, GRK2, as well as arrestin3 to muscarinic acetylcholine $M_3$ receptors ($M_3R$) as a model. These effector proteins and their recruitment were monitored by means of single-cell Förster resonance energy transfer (FRET) imaging under conditions specifically optimized to minimize interference of upstream events (G protein activation in case of GRK2 and prephosphorylation in case of arrestin). By investigating the effect of seven different agonists on these effectors, we answered the following two questions: firstly, can we detect biased recruitment of effector proteins at the $M_3R$ between these seven agonists? Secondly, is a potential bias in arrestin3 recruitment caused by a bias in GRK2 recruitment or can both arrestin3 and GRK2 recruitment be biased differently in comparison to $G_q$ binding and activation?

Our results show that ligand-induced biased recruitment can indeed be detected at the $M_3R$. Assessing bias as a change in sequence when ranking the agonists with respect to their efficacy to induce $G_q$, GRK2, or arrestin3 recruitment, we found a difference between arrestin3 recruitment and the other two effector proteins. In contrast, no such difference was found between GRK2 recruitment and G protein binding and activation, suggesting that very similar active receptor conformations are required for $G_q$ activation and GRK2 recruitment.

## Results
### Assays measuring agonist bias at the $M_3R$

We selected seven agonists with pharmacological relevance or structural similarity to a pharmacologically relevant agonist (*Figure 1—figure supplement 2*): acetylcholine (ACh), arecoline (Are), methacholine (Metha), pilocarpine (Pilo), guvacoline (Guva), methyl-3-dimethylaminopropionate (Mda), and 5-methyl-furmethiodide (Fur). The ability of every agonist to enable $G_q$, GRK2, or arrestin3 binding to the $M_3R$ was investigated with single-cell FRET-based measurements in human embryonic kidney 293T (HEK293T) cells. As expected, receptor stimulation with 30 µM ACh resulted in a reversible increase in YFP fluorescence and a corresponding decrease in mTurq fluorescence, reflecting the FRET development due to $G_q$, GRK2, or arrestin3 binding to the $M_3R$, respectively (*Figure 1—figure supplement 3*). Thus, we were able to compare multiple agonists in the same cell under the same conditions.

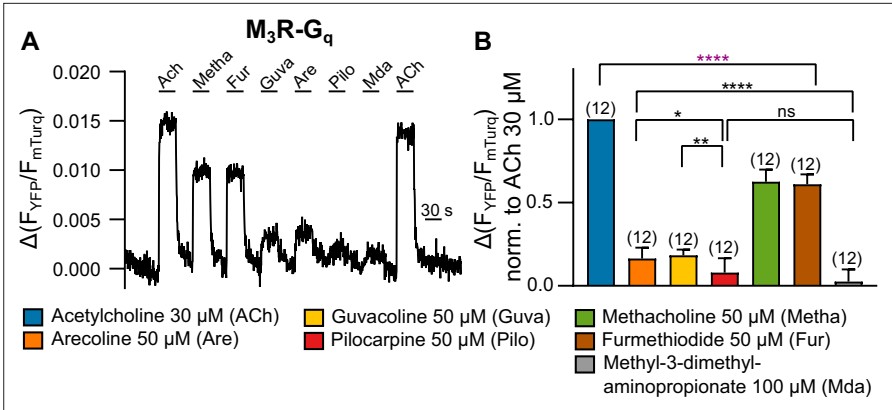

**Figure 1.** Measuring G protein binding to the M₃R upon stimulation with muscarinic receptor agonists (**A**). (**B**) G protein-M₃R interaction was measured as described in *Figure 1—figure supplement 3* by means of single-cell Förster resonance energy transfer (FRET) recording. The cells were stimulated with distinct concentrations of each agonist as indicated. The amplitudes were normalized to the ACh amplitude of every cell. An ordinary one-way ANOVA (****p<0.0001) with Tukey's multiple comparison test, and additionally for paired measurements a paired Student's *t*-test, was conducted. Only the significant differences (ANOVA: black, *t*-test: purple) in terms of biased recruitment to the M₃R are shown. All data are plotted as mean values ± SD for each condition.

The online version of this article includes the following source data and figure supplement(s) for figure 1:

**Source data 1.** Source data related to *Figure 1B*.

**Figure supplement 1.** Schematic representation of G protein-coupled receptor (GPCR) activation: agonist binding to the orthosteric pocket stabilizes the active receptor conformations.

**Figure supplement 2.** 2D chemical structures of the seven agonists.

**Figure supplement 3.** Measuring response modulation of muscarinic receptor agonists of Gq, GRK2, and arrestin3 binding to the M₃ muscarinic acetylcholine receptor (M₃R).

**Figure supplement 4.** Measuring Gq activation of muscarinic receptor agonists.

**Figure supplement 4—source data 1.** Source data related to *Figure 1—figure supplement 4*, concentration-response curves of Gq activation.

**Figure supplement 5.** Measuring affinity of seven muscarinic receptor agonists at the M₃R.

**Figure supplement 5—source data 1.** Source data related to *Figure 1—figure supplement 5*.

## Agonist efficacy is reflected in the amplitudes of M₃R-Gq protein binding

For many receptors including M₃R, it is well known that signal amplification at the level of G protein activation leads to a spare receptor phenomenon, which allows only indirect determination of agonist efficacy by means of comparing of concentration-response curves with those of ligand binding. Therefore, we determined agonist evoked binding of Gq to M₃R by means of single-cell FRET recording as a direct measurement of agonist efficacy (*Figure 1A and B*). During laminar superfusion of cells, we sequentially switched between buffer and all seven agonists used in the study. Due to fast kinetics, we were able to wash out the respective agonists before we added the next. In order to get a robust assessment of agonist efficacy, we used high agonist concentrations for all agonists and kept them constant for comparison in the GRK2 and arrrestin3 recruitment assays. At the end of the protocol, we reapplied 30 µM ACh in order to test for the stability of the signal. Our results showed the highest recruitment of Gq for ACh, followed by Metha and Fur (~60% of the ACh signal), and much weaker recruitment for Guva and Are (~20%) followed by even lower values for Pilo and Mda (~8%) (*Figure 1B*), leading to the following ranking in efficacy: ACh>Metha≈Fur>Are≈Guva>Pilo>Mda. We also performed bioluminescence resonance energy transfer (BRET)-based assays to study Gq activation by these agonists (*Figure 1—figure supplement 4*), with the following similar ranking in potency: ACh>Metha≈Fur>Are>Guva>Pilo>Mda. In addition, radioligand displacement assays were performed (*Figure 1—figure supplement 5*) in order to determine the rank order of agonist affinity: Pilo>ACh>Metha≈Fur>Are>Guva»Mda. Mda failed to displace the radioligand at concentrations up

to 100 mM, indicating a very low affinity for the orthosteric binding pocket of M$_3$R or – despite its structural similarity to ACh – a noncompetitive binding mode.

## G protein-independent GRK2 recruitment to the M$_3$R

Our aim to measure and compare the efficacy of the agonists also at the level of GRK2 recruitment to M$_3$R was complicated by the fact that GRK2 is activated by G protein-induced recruitment to the plasma membrane. The rank order of the efficacy of the different agonists to recruit GRK2 to M$_3$R, determined by single-cell FRET assays, was the same as determined for G$_q$ recruitment (*Figure 2—figure supplement 1A and B* in comparison with *Figure 1A and B*). However, due to possible interference with G$_q$ in the recruitment of M$_3$R, these data are difficult to interpret. Indeed, when G$_q$ activation was completely abolished by pretreatment with FR900359, no recruitment of GRK2 to M$_3$R was observed in cells transfected with G$_q$ and fluorescently labeled M$_3$R and GRK2 (*Figure 2—figure supplement 1C*). Likewise, preincubation with PTX completely abolished agonist-evoked FRET between G$_{\beta\gamma}$ and GRK2 (*Figure 2—figure supplement 1D*). Interestingly, without overexpression of G$\alpha_q$ proteins, preincubation with PTX and FR900359 pretreatment did not lead to a full inhibition of agonist evoked FRET between G$_{\beta\gamma}$ and GRK2 (*Figure 2—figure supplement 1E*), presumably due to bystander FRET upon G protein-independent translocation of GRK2 to M$_3$R. Kinetic analysis of agonist-evoked FRET between G$_{\beta\gamma}$ and GRK2 with or without pretreatment with PTX and FR900359 showed fast one-phased decays for signals measured with pretreatment and slower two-phased decays for signals measured without pretreatment (*Figure 2—figure supplement 1F and G*). In order to abolish the possible involvement of G$_{\beta\gamma}$ and G$\alpha_q$ proteins in the recruitment process of GRK2 to M$_3$R (*Wolters et al., 2015*) further, we aimed to bypass G$_q$-dependent translocation of GRK2 to the plasma membrane by introducing a CAAX-box at the C-terminus of GRK2-YFP to allow for G protein-independent plasma membrane localization of GRK2 (*Inglese et al., 1992*). In addition, we mutated D110 to alanine, which attenuates G$\alpha_q$ binding to GRK2 (*Wolters et al., 2015*). Under conditions of complete G$_q$ and G$_i$ inhibition (see for control *Figure 2—figure supplement 1D and E*), amplitudes of agonist-evoked FRET increases were reduced and offset kinetics after agonist withdrawal were substantially faster (compare *Figure 2A* and *Figure 2—figure supplement 1A*). However, we were still able to measure agonist-specific interactions between (D110A)GRK2-CAAX and M$_3$R (*Figure 2A and B*, *Figure 2—figure supplement 1H*). Our measurements of the relative efficacy of the agonists to recruit this GRK2 construct without the influence of G proteins resulted in the following ranking: ACh>Metha≈Fur>Guva≈Are>Pilo (*Figure 2B*). Mda did not lead to detectable responses. Even though the relative responses induced by the weak partial agonists were substantially reduced compared to the G protein-dependent recruitment of GRK2 (*Figure 2—figure supplement 1A and B*), the rank order of the agonists was essentially unchanged. We ensured that the color switch of the attached fluorophores had no effect on the relative agonist efficacies (*Figure 2—figure supplement 1I*). Control experiments without overexpression of G$\alpha_q$, but in the context of non-mutated fluorescent GRK2, showed a similar efficacy of the partial agonists Pilo and Are for GRK2 binding to M$_3$R compared to those measured under complete G protein inhibition with the G protein-insensitive GRK2 (*Figure 2—figure supplement 1J and K*). These results demonstrate, as previously reported (*Wolters et al., 2015*), that the overexpression of G$_q$ proteins leads to an enhanced affinity of GKR2 for the M$_3$R.

## Agonist-induced arrestin3 recruitment to prestimulated M$_3$R

Previous studies have shown that GRK2-mediated phosphorylation of the receptor is initially the rate-limiting step of arrestin binding and that β2-adrenergic receptor-arrestin interaction and M$_3$R-arrestin interaction accelerate with repeated stimulation of the same cell (*Krasel et al., 2005*; *Wolters et al., 2015*). Since GRK-mediated receptor phosphorylation is in most cases a prerequisite for agonist-evoked arrestin3 binding (*Choi et al., 2018*), it is difficult to selectively measure the efficacy of agonists to induce binding of arrestin3 to (phosphorylated) receptors. The dependency on agonist-induced phosphorylation for arrestin binding is exemplarily shown in *Figure 2C and D* as Metha-induced arrestin3 recruitment is significantly enhanced after a prepulse with the full agonist ACh.

To minimize the influence on arrestin3 recruitment by each agonist's potentially different ability to activate G proteins and subsequently GRK2, we analyzed the time dependency and GRK2 dependency of the arrestin3 recruitment (*Figure 2—figure supplement 2A–G*) and developed a prestimulation

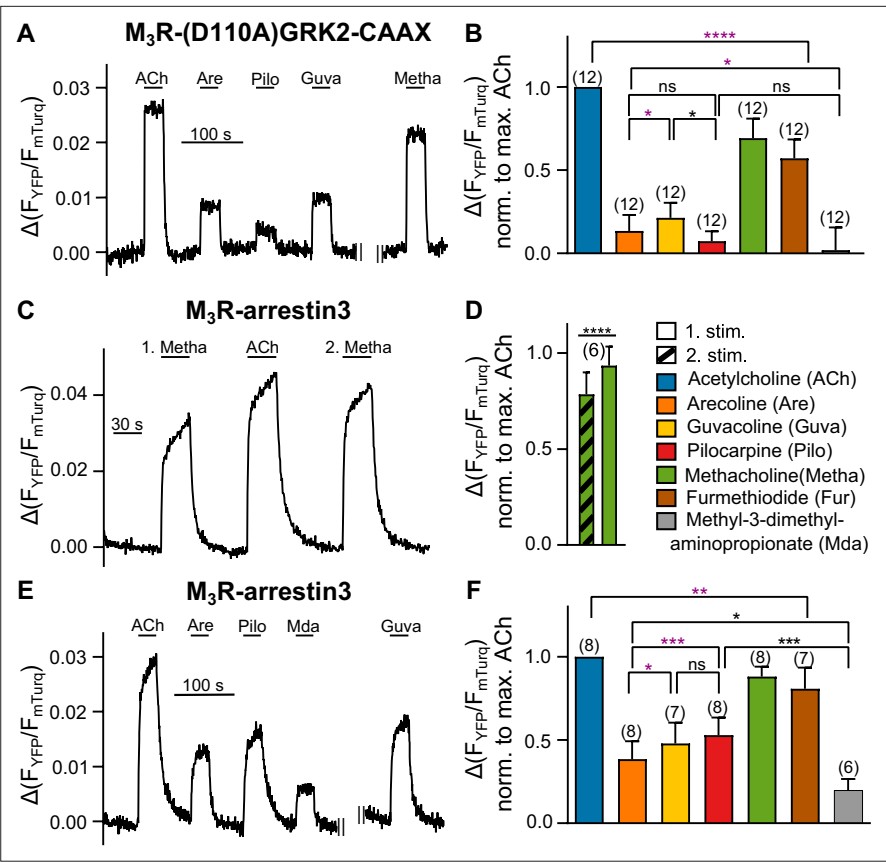

**Figure 2.** Bias in arrestin3 recruitment is distinct from bias in GRK2 recruitment. (**A, B**) M$_3$R-(D110A)GRK2-CAAX interaction without Gα$_q$ overexpression was measured with a single-cell Förster resonance energy transfer (FRET)-based assay, after cells were preincubated with 50 ng/mL of PTX overnight and additionally for 10 min before the start of the measurement, while continuously superfused with FR900359 at a concentration of 1 μM during the whole measurement. (**C–F**) M$_3$R-arrestin3 interaction was measured with a single-cell FRET-based assay. Each cell was stimulated for 30 s with saturating concentrations of each indicated agonist. (**C**) Stimulation with agonist (first stimulation), thereafter with ACh and after that with agonist again (second stimulation) shows that M$_3$R-arrestin3 interaction increases with repeated stimulation by an agonist in the same cell. (**D**) Mean amplitude values of first and second stimulation were statistically analyzed with a paired $t$-test. (**B, D, F**) All amplitudes were normalized to the ACh amplitude of the same cell (norm. to max. ACh). One-way ANOVA (****$p<0.0001$) with Tukey's multiple comparison test, and additionally for paired measurements a paired Student's $t$-test, was conducted. Only the significant differences (ANOVA: black, $t$-test: purple) in terms of biased recruitment to the M$_3$R are shown. Data are shown as mean ± SD, *$p<0.05$, **$p<0.001$, ***$p<0.001$, ****$p<0.0001$. The number of experiments is indicated in parentheses.

The online version of this article includes the following source data and figure supplement(s) for figure 2:

**Source data 1.** Source data related to *Figure 2B, D and F*.

**Figure supplement 1.** Förster resonance energy transfer (FRET)-based single-cell measurements to control successful eradication of the influence of G protein activation on M$_3$R-GRK2 interaction.

**Figure supplement 1—source data 1.** Source data related to *Figure 2—figure supplement 1B, D, E, F, F, I and K*.

**Figure supplement 2.** M$_3$R-arrestin3 interaction increases with repeated stimulation by an agonist in the same cell.

**Figure supplement 2—source data 1.** Source data related to *Figure 2—figure supplement 2B, D, E and G*.

**Figure supplement 3.** The sequence of measuring M$_3$R-arrestin3 interaction for different agonists in the same cell does not influence the results.

**Figure supplement 3—source data 1.** Source data related to *Figure 2—figure supplement 3*.

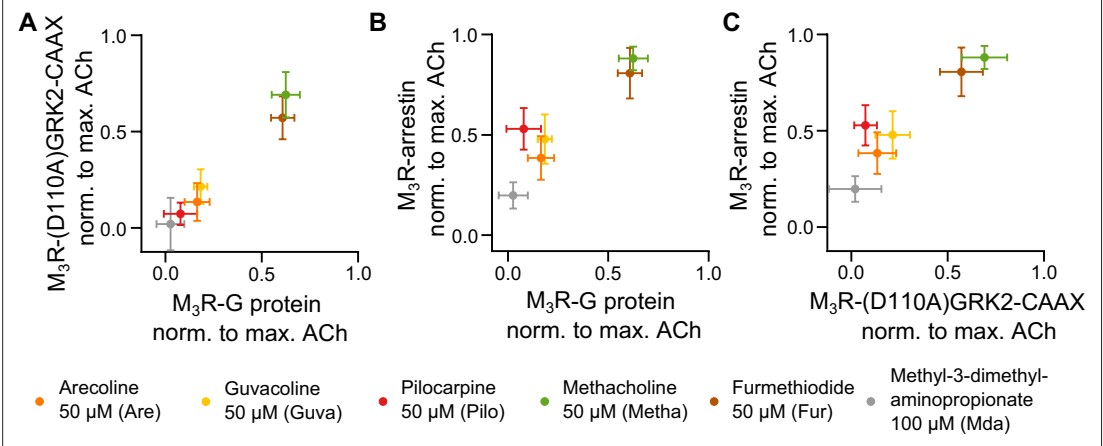

**Figure 3.** Measuring GRK2 recruitment disentangled from $G_q$ reveals no bias between $G_q$ and GRK2 recruitment to $M_3R$. Recruitment of $G_q$ (data from **Figure 1**) was plotted in relation to (**A**) recruitment of (D110A)GRK2-CAAX and (**B**) recruitment of arrestin3 (data from **Figure 2**) for six agonists normalized to the maximum amplitude of ACh at a concentration of 30 µM. (**C**) The normalized $M_3R$-arrestin3 recruitment was plotted in relation to the $M_3R$-(D110A)GRK2-CAAX recruitment.

The online version of this article includes the following source data and figure supplement(s) for figure 3:

**Figure supplement 1.** Förster resonance energy transfer (FRET)-based single-cell measurement of maximum response for Fur and Mda for (**A**) $M_3R$-arrestin3 and (**B**) $M_3R$-GRK2 interaction.

**Figure supplement 1—source data 1.** Source data related to **Figure 3—figure supplement 1A, B**.

**Figure supplement 2.** Identification of methacholine, pilocarpine, and arecoline as biased agonists at the $M_3R$ relative to ACh.

**Figure supplement 2—source data 1.** Source data related to **Figure 3—figure supplement 2A**.

**Figure supplement 2—source data 2.** Source data related to **Figure 3—figure supplement 2B**.

**Figure supplement 2—source data 3.** Source data related to **Figure 3—figure supplement 2C**.

protocol. The amplitudes measured for prestimulated and not prestimulated $M_3R$ differed for all seven agonists except for ACh (**Figure 2—figure supplement 2D**). This indicates that the increase in amplitude after prestimulation is specific for each agonist (**Figure 2—figure supplement 2E**), presumably due to differences in their capability to induce phosphorylation of $M_3R$ (**Butcher et al., 2011**). The prestimulation protocol contained a first stimulation with saturating concentrations of ACh, followed by a brief washout, immediately followed by test pulses (**Figure 2E**). We verified the reliability of the protocol by comparing relative FRET amplitudes, evoked by the test compounds when applied in different order, with the amplitudes obtained directly after prestimulation with ACh (**Figure 2—figure supplement 3**). By applying each of the seven agonists to the $M_3R$ subsequent to the prestimulation with ACh, we obtained results that show marked differences in the ability of the individual agonists to recruit arrestin3 to $M_3R$ (**Figure 2E and F**). This led to the following efficacy ranking of the agonists: ACh>Metha≈Fur>Pilo≈Guva>Are>Mda. It is important to note that the rank order for the efficacy to recruit arrestin3 to $M_3R$ differs in comparison to the one obtained for $G_q$ recruitment and activation.

## Agonist-induced $M_3R$-arrestin3 recruitment can be different from $M_3R$-$G_q$ and $M_3R$-GRK2 recruitment

In order to compare the relative efficacies (normalized to ACh) of all agonists tested for binding of the three different effector proteins to $M_3R$ as determined from experiments shown in **Figures 1B, 2B and F**, we used 2D plots and compared all three modalities pairwise, as depicted in **Figure 3A–C**. As shown in **Figure 3A**, the agonist efficacies to recruit $G_q$ to $M_3R$ in comparison to D110A-GRK-CAAX were very similar, showing nearly equal efficacy for each agonist. In contrast, both 2D plots comparing agonist efficacies for arrestin3 binding to $M_3R$ with those for either $G_q$ or D110A-GRK2-CAAX showed higher relative agonist efficacies for arrestin binding. Most importantly, the rank order of agonist efficacies was different for arrestin3 binding as Pilo was more efficient than Are and similarly efficient as Guva to induce arrestin3 recruitment, whereas for both GRK2 recruitment and $G_q$ binding to $M_3R$, Pilo was less efficient than Are and Guva. We further determined efficacies

of the agonists with respect to G protein activation (*Figure 1—figure supplement 4*), GRK2 recruitment with $G_q$ expression and without G protein inhibitors (*Figure 3—figure supplement 1B*, *Figure 3—figure supplement 2*) as well as arrestin3 recruitment (*Figure 3—figure supplement 1A*, *Figure 3—figure supplement 2*). Using the determined binding affinities for the agonists (*Figure 1—figure supplement 5*), the obtained curves were fitted according to the operational model of *Black and Leff, 1997*, yielding their respective $\tau$ values. The $\Delta log(\frac{\tau}{K_i})$ values of the selected agonists relative to ACh in all three assays were then plotted in a radar plot (*Figure 3—figure supplement 2*). The results confirmed that agonist efficacies for GRK2 recruitment and $G_q$ activation followed a similar pattern with the identical rank order of agonists, whereas in the case of arrestin3 binding, at least Pilo exhibited a reverted rank order relative to Are and Guva.

## Discussion

The notion that different agonists can stabilize different conformational states of the receptor is well accepted within the field of GPCR pharmacology, and numerous studies indicate that selective activation of signaling pathways might at least in part be induced by these different conformations (*Wingler et al., 2019*). So far, a direct comparison of the bias of agonists at the level of binding of G proteins, GRKs, and arrestins to receptors has been lacking. Instead of analyzing downstream signaling, we directly investigated the recruitment of $G_q$, GRK2, and arrestin3 to the $M_3R$ using FRET-based approaches. The specific aim and, at the same time, difficulty was to analyze these events, which are ultimately interconnected, in a mutually independent way in order to differentiate between their conformational requirements. Specifically, there are two main dependencies that need special attention: (1) recruitment of GRK2 to the plasma membrane is enhanced by $G\alpha_q$ and $G_{\beta\gamma}$ subunits upon G protein activation; and (2) arrestin binding to receptors requires both GRK-mediated phosphorylation of receptors and receptor activation. Our experimental design accounted for both processes.

(1) We measured GRK2 recruitment under conditions of complete $G_{i/o}$ and $G_q$ inhibition by preincubation with PTX and FR900359 (*Figure 2A and B*). To circumvent G protein-mediated membrane recruitment of GRK2, we included a CAAX-motif at the C-terminus of the fluorophore to induce isoprenylation (*Inglese et al., 1992*), and in addition a D110A mutation to attenuate interaction with $G\alpha_q$ (*Wolters et al., 2015*). However, we had to omit coexpression of $G_q$ in FR900359-treated cells, otherwise the interaction between $M_3R$ and GRK2 was completely blocked (*Figure 2—figure supplement 1C*), presumably due to occupation of receptors by inactivated G proteins, and thereby blockade of the interaction site for GRK2 binding. Strikingly, the rank order of agonist efficacies to recruit GRK2 was the same, independent of whether G proteins were inhibited or not. However, the overall FRET amplitudes and also the agonist responses normalized to the one of ACh were considerably larger and their overall kinetics considerably slower in the absence of G protein inhibition, demonstrating the contribution of G proteins to translocating GRK2 to the plasma membrane. Importantly, the rank order of the efficacy of Metha, Are, and Pilo to recruit GRK2 to the $M_3R$ (*Figure 2A and B*, *Figure 2—figure supplement 1A and B*) is the same as observed for $M_3R$ phosphorylation determined by Butcher et al. (Figure 6 in *Butcher et al., 2011*), indicating that GRK2 recruitment to the $M_3R$ correlates with functional phosphorylation.

(2) To avoid an influence of the different efficacies of agonists when recruiting GRK2 and thus the receptor being phosphorylated, we established a specific single-cell FRET protocol. We confirmed arrestin recruitment to the $M_3R$ to be dependent on GRK2-mediated phosphorylation (*Figure 2—figure supplement 2*), as in the absence of coexpression of GRK2 or upon expression of the dominant negative GRK2(K220R), a decrease of the arrestin recruitment was detected as expected based on previous observations at the $\beta_2$-adrenergic receptor (*Krasel et al., 2005*). All cells were preexposed to the full agonist ACh for a defined period of time, in order to allow for phosphorylation of receptors before ACh was withdrawn, followed by the measurement of arrestin recruitment induced by the seven different agonists (*Figure 2E and F*) to the thus prephosphorylated receptors. As the dephosphorylation of receptors typically occurs on the order of at least several minutes, our experimental design allowed the comparison of agonist efficacies to prephosphorylated receptors, minimizing any influence of agonist efficacy towards G protein activation and GRK2 binding. Our results show that unlike for G protein and GRK2 recruitment, Pilo induces significantly more arrestin3 recruitment compared to Are. These results are in line with a potential bias of Pilo

towards arrestin that was reported by *Pronin et al., 2017*. However, here we were able to observe that the bias towards arrestin recruitment was not introduced by a bias towards GRK2 recruitment. In contrast, no detectable bias was observed between $G_q$ recruitment and GRK2 recruitment for any of these agonists. Our findings disentangle bias between $G_q$, GRK2, and arrestin3 recruitment to the $M_3R$. Arrestin3 recruitment was tested in an overexpression system for all agonists under exactly the same conditions regarding prephosphorylation and kinase expression. The influence of endogenous GRK and arrestin isoforms is therefore negligible. However, we cannot rule out that the overall agonist-dependent recruitment might as well depend on the expression pattern of GRKs. Since $M_3R$-mediated signaling is regulated by distinct mechanisms (*Luo et al., 2008*), our results represent the first step on the way to completely disentangle bias signaling. Measurement of agonist-induced G protein recruitment to receptors by means of FRET is a much more direct way to study the efficacy of agonists to activate receptors compared to the measurement of G protein activation or downstream signaling due to its lack of signal amplification, which typically leads to the phenomenon of signal saturation at submaximal levels of receptor activation. However, our determination of the efficacy of the different agonists to activate G proteins by calculating $\Delta log(\frac{\tau}{K_i})$ resulted in the same rank order for the agonists, confirming the applicability of receptor G protein interaction assay. We determined also saturating agonist concentration for our GRK2 and arrestin3 recruitment measurements. These concentrations were used for our measurement of agonist efficacies, and, to ensure comparable conditions, to measure G protein binding as well. Having taken very serious care of measuring all three binding events completely independent of each other, a comparison of the agonist efficacies for all three events was possible. Metha and Fur recruited arrestin3, GRK2, and $G_q$ to a similar extent compared to the full agonist ACh, whereas the partial agonists Are, Guva, and Pilo differed substantially in their relative efficacy to recruit arrestin3 versus GRK2 and $G_q$ (*Figures 1B, 2B and F*). The strongest deviation in the rank order was observed for Pilo, which from the set of partial agonists was the poorest GRK2 and $G_q$ recruiter, but the best arrestin3 recruiter, indicating a relative bias towards arrestin3 binding. Mda did not show a detectable effect in the radioligand replacement assay and in the recruitment of (D110A)GRK2-CAAX. Furthermore, it showed only very low levels of recruitment with respect to $G_q$, low level of recruitment with respect to arrestin3, and a medium level of $G_q$ activation and GRK2 recruitment in the presence of intact $G_q$. Therefore, it is difficult to ascertain its precise mechanism of action. Even though we did not directly measure GRK2-mediated phosphorylation, we found that the slow component of arrestin3 recruitment upon first stimulation with agonist was dependent on the expression of catalytically active GRK2, similar to what has been described for $\beta_2$-adrenergic receptors (*Krasel et al., 2005*).

However, this slow component, even though it is smaller in amplitude, was also visible during subsequent applications of agonist. This could either indicate incomplete phosphorylation at the time points of the repetitive agonist applications or could reflect phosphorylation-independent effects, such as the interaction of arrestin3 with the membrane and a thus prolonged residence time of arrestin3 at the membrane after the first recruitment to the receptor (*Lally et al., 2017*). Our finding that agonist bias with respect to the ability to activate G proteins versus the ability to induce arrestin recruitment to the receptor can be detected correlates nicely with recent advancements in the determinnation of conformational differences of the angiotensin-II receptor type 1 when bound to $G_q$ or arrestin (*Wingler et al., 2020*; *Suomivuori et al., 2020*). Furthermore, our results demonstrating the existence of functional agonist-dependent patterns of differential recruitment of arrestin3 and GRK2 to the $M_3R$ are supported by a recent study demonstrating agonist-dependent patterns of recruitment of G proteins, GRK2, and a conformationally selective nanobody directed against the intracellular cavity of activated opioid receptors (*Stoeber et al., 2020*). The finding that different receptor ligands can differentially affect binding of G proteins, arrestins, and GRKs opens up the potential for future drug development to specifically direct signaling in one or the other direction. In summary, our findings highlight the existence of ligand-induced bias at the $M_3R$ and the importance of understanding GRK2 recruitment and its role for the subsequent arrestin3 recruitment in order to fully differentiate between a bias in GRK2 recruitment and a bias in arrestin3 recruitment. While developed for the $M_3R$, the methodology is generally applicable to all GPCRs, G proteins, GRKs, and arrestins, and therefore offers new possibilities to disentangle biased effector pathway engagement at the level of effector protein recruitment and activation.

# Materials and methods

**Key resources table**

| Reagent type (species) or resource | Designation | Source or reference | Identifiers | Additional information |
|---|---|---|---|---|
| Chemical compound, drug | Acetylcholine iodide | Sigma-Aldrich | CAS number: 2260-50-6 | |
| Chemical compound, drug | Arecoline hydrobromide | TCI Chemicals | CAS number: 300-08-3 | |
| Chemical compound, drug | Methacholine chloride | TCI Chemicals | CAS number: 62-51-1 | |
| Chemical compound, drug | Guvacoline hydrobromide | TRC Canada | CAS number: 17210-51-4 | |
| Chemical compound, drug | Methyl-3-(dimethylamino) propionate | Sigma-Aldrich | CAS number: 3853-06-3 | |
| Chemical compound, drug | Pilocarpine hydrochloride | TCI Chemicals | CAS number: 54-71-7 | |
| Chemical compound, drug | Trimethyl-(5-methyl-furan-2-ylmethyl)-ammonium iodide | Sigma-Aldrich | CAS number: 1197-60-0 | |

## Reagents

Coelenterazine h was obtained from NanoLight, Pinetop, AZ. Dulbecco's Modified Eagle's Medium (DMEM), PBS, penicillin/streptomycin, and Trypsin-EDTA were from Capricorn Scientific Gmbh, Ebsdorfergrund, Germany. Poly-L-lysine hydrobromide, FCS, L-glutamine, PEI, acetlycholine iodide, methyl-3-(dimethylamino)propionate (Mda), and trimethyl-(5-methyl-furan-2-ylmethyl)-ammonium iodide (Fur) were obtained from Sigma-Aldrich, Merck KGaA, Darmstadt, Germany. Arecoline hydro-bromide, pilocarpine hydrochloride, and methacholine chloride were from TCI Chemicals, Eschborn, Germany, and guvacoline hydrobromide from TRC Canada, Toronto, Canada. Pertussis toxin was purchased from EMD Millipore Corp., Merck KGaA. FR900359 was isolated from *Ardisia crenata* leaves as previously described in *Schrage et al., 2015*.

## Plasmids

cDNAs for $G\alpha_q$, $G\alpha_q$-yellow fluorescent protein (YFP), where YFP was inserted between residues 124 and 125 of $G\alpha_q$ (*Hughes et al., 2001*), $G\beta_1$, $G\gamma_2$ (*Bünemann et al., 2003*), $G\alpha_o$, mTurquoise2-$G\gamma_2$, where mTurquoise2 was fused to the N-terminus of $G\gamma_2$ (*Jelinek et al., 2021*), $M_3$R-YFP, where YFP was fused C-terminally to $M_3$R (*Hoffmann et al., 2012*), GRK2 (*Winstel et al., 1996*), GRK2-mTurquoise, and GRK2-YFP, where mTurquoise or YFP were fused to the GRK2 C-terminus (*Wolters et al., 2015*), and arrestin3-mTurquoise, where mTurquoise was fused to the C-terminus of arrestin3 (*Krasel et al., 2005*; *Miess et al., 2018*; *Roseberry et al., 2001*), were described previously. $M_3$R-mTurquoise was cloned by an exchange of mCit to mTurquoise of $M_3$R-mCit (*Jelinek et al., 2021*), where mTurquoise was fused at the C-terminus. The cDNA for the $M_3$R was obtained from the Missouri S&T cDNA Resource Center and pcDNA3 from Invitrogen. cDNA containing pNluc was kindly provided by Dr. N. Lambert (Augusta Medical College, GA), and pNluc-$G\gamma_2$, where Nluc was fused to the N-Terminus of G$G\gamma_2$, was kindly provided by Dr. C. Krasel (Philipps-Universität Marburg, Germany). GRK2-YFP-CAAX was cloned by amplifying the cDNA for YFP or mTurquoise by PCR with the forward primer AAA AAA TCT AGA GTG AGC AAG GGC GAG G and the reverse primerAAAAAAGCGGCCGCCTAgga-gagcacacacttgcagctcatgcagcccgggccactctcatcaggagggttCTTGTACAGCTCGTCCATGC. The reverse primer attaches the last 18 amino acids of H-Ras to the C-terminus of the fluorescent protein. The resulting PCR product was digested with XbaI and NotI and cloned into pcGRK2-YFP (*Wolters et al., 2015*) that had been digested with the same enzymes, replacing the YFP with the modified fluores-cent protein. (D110A)-GRK2-YFP-CAAX was generated from GRK2-YFP-CAAX by mutagenesis analo-gously to (D110)-GRK2-YFP in *Wolters et al., 2015* using the primer: CCG GGA GAT CTT CGC CTC ATA CAT CAT G.

## Cell culture and transfection

For experiments HEK tsA 201 cell line was used, which was a kind gift from the Lohse laboratory, University of Würzburg. Therefore, all experiments were carried out in HEK293T cells. They were cultured in DMEM with high glucose and supplemented with 10% FCS, 2 mM L-glutamine, 100 U/mL penicillin, and 0.1 mg/mL streptomycin. Cells were transiently transfected 24 hr after seeding (6 cm

dish) with linear polyethylenimine (PEI) 25 kDa as transfecting agent. For $M_3R$-G protein interaction experiments, cells were transfected with cDNAs as follows: 1.5 µg $M_3R$ -YFP, 2.4 µg $G\alpha_q$, 0.75 µg $G\beta_1$, and 0.3 µg mTurquoise2-$G\gamma_2$. For $M_3R$ GRK2 interaction experiments, cells were transfected with the following cDNAs: 1.5 µg $M_3R$-YFP, 2.4 µg $G\alpha_q$, 0.75 µg $G\beta_1$, 0.3 µg $G\gamma_2$, and 0.75 µg GRK2-mTurquoise. For $M_3R$ (D110A)-GRK2-YFP-CAAX interaction experiments, the cells were transfected with 1.5 µg $M_3R$-mTurquoise, 2.4 µg pcDNA3, 0.75 µg $G\beta_1$, 0.3 µg $G\gamma_2$, and 0.75 µg (D110A)-GRK2-YFP-CAAX. For $M_3R$ GRK2 interaction experiments without $G\alpha_q$ overexpression, the cells were transfected with 1.5 µg $M_3R$-mTurquoise, 2.4 µg pcDNA3, 0.75 µg $G\beta_1$, 0.3 µg $G\gamma_2$, and 0.75 µg GRK2-YFP. For $M_3R$-arrestin interaction experiments, cells were transfected with the following amounts of cDNAs: 1.5 µg $M_3R$-YFP, 0.75 µg GRK2, and 1.5 µg arrestin3-mTurquoise. For $G\beta_1$/$G\gamma_2$ GRK2 interaction experiments, the cells were transfected with: 1.5 µg $M_3R$, 2.4 µg pcDNA3, 0.75 µg $G\beta_1$, 0.3 µg mTurquoise2-$G\gamma_2$, and 0.75 µg GRK2-YFP and at the $M_2R$ with 1.5 µg $M_2R$ 2.4 µg $G\alpha_o$, 0.75 µg $G\beta_1$, 0.3 µg mTurquoise2-$G\gamma_2$, and 0.75 µg GRK2-YFP. For BRET-based $G\alpha_q$ activation experiments, cells were transfected with cDNAs as follows: 1.5 µg $M_3R$, 2.4 µg $G\alpha_q$-YFP, 0.75 µg $G\beta_1$, 0.3 µg pNuc-$G\gamma_2$, and 0.75 µg GRK2. The mixing ratio of PEI to DNA was 3:1. Per 1 µg DNA, 50 µL DMEM without FCS were added to the DNA and PEI solutions, respectively. Both solutions were mixed, added together and incubated, protected against light, at 20°C for 30 min. The mix was added to the cells and incubated at 37°C in a humidified atmosphere of 95% air and 5% $CO_2$. For FRET-based experiments, cells were seeded onto 6-well plates with 25 mm coverslips coated with poly-L-lysine after 24 hr. For BRET-based experiments, cells were counted after 24 hr and 16,000 cells/well were seeded in 96-well plates (Greiner 96 Flat White) coated with poly-L-lysine. After 48 hr of transfection, measurements were performed at room temperature.

## Single-cell FRET-based measurements

Unless indicated otherwise, FRET-based measurements on single cells were performed as described in *Milde et al., 2013*. Transiently transfected cells were subjected to single-cell time-resolved FRET imaging with constant superfusion of either buffer (137 mM NaCl, 5.4 mM KCl, 2 mM $CaCl_2$, 1 mM $MgCl_2$, 10 mM HEPES, pH 7.3) or buffer-containing agonist. Cells were measured with an inverted fluorescence microscope (Eclipse Ti; Nikon, Düsseldorf, Germany). Light-emitting diodes (LED) at 425 nm and 500 nm were used for excitation. The intensity of both LEDs of the excitation system (pE-2; CoolLED, Andover, UK) was set to 2%. Fluorescence intensity was measured using the imaging software NIS-Elements advanced research (Nikon Corporation) and recorded at 2 Hz. The FRET ratio was calculated as the fluorescence intensity ratio of YFP and mTurquoise emission after mTurquoise excitation at 425 nm. Stimulation with agonist led to an increase in fluorescence intensity ratio, reflecting the interaction of the $M_3R$ with arrestin3 or GRK2. The fluorescence data were corrected for background fluorescence, bleed-through, and false excitation, and then plotted over time. The presented data were baseline-corrected to account for photobleaching. Time intervals of every measurement are indicated in the corresponding figure legend. For averaging the single FRET-based measurements, each measurement was normalized to the induced individual maximum response of saturating concentrations of ACh (30 µM concentration) if not indicated otherwise. Amplitudes were determined by calculating the difference between the mean of FRET ratio values 5 s before stimulation with the agonist and 5 s before withdrawal of the agonist. For FRET-based measurements analyzing the increase of receptor-arrestin3 interaction upon repeated stimulation by the same agonist, each cell was stimulated with agonist for 30 s, followed by a stimulation with ACh to make sure that the maximum of the effect was reached. Afterward, another stimulation of 30 s with agonist followed, termed the second stimulation. Pilo and ACh were measured as additional controls in every cell, and the amplitude of the stimulation with ACh was used for normalization. The normalized $M_3R$ -arrestin3 interaction level of each agonist was compared between first and second stimulation. For FRET-based experiments using pertussis toxin for $G\alpha_{i/o}$ inhibition, the cells were pretreated overnight with a resulting concentration of 50 ng/mL PTX. For inhibition of $G\alpha_q$ using FR900359, the cells were preincubated 10 min before the start of the measurement and superfused during the whole measurement as buffer and agonist solutions contained 1 µM FR900359.

## BRET-based measurement

Bioluminescence resonance energy transfer (BRET)-based measurements were conducted with the luciferase reporter Nluc (*Hall et al., 2012*). Transiently transfected adherent cells were measured with a Spark 20M Multimode Microplate Reader from Tecan. Fluorescence and luminescence intensities were obtained using the SparkControl application (Tecan). Cells were carefully washed with buffer (as described in single-cell FRET-based measurements). Afterward, coelenterazine h in buffer was added to the cells. Every well contained a volume of 80 µL and a final concentration of 3.07 µM coelenterazine h. The BRET ratio was calculated as the quotient of the YFP signal (light emitted between 520 nm and 700 nm) and the nanoLuc signal (light emitted between 415 nm and 485 nm). After 10 min incubation with coelenterazine h, measurement of the baseline BRET ratio took place for 10 cycles (*baseline phase*). One cycle had a duration of about 44 s. After 10 cycles, this led to a total of 6.5 min of baseline measurement. The measurement was paused and 20 µL buffer (negative control) or agonist was added to the cells. The development of BRET was measured for 10 cycles (*agonist phase*) and then paused again for the addition of 20 µL of a solution containing ACh in buffer. The final concentration of ACh in each well was at least 50 µM, aimed at creating saturating conditions and measuring the maximum response obtainable in each individual well. Afterward, 10 more cycles were measured (*saturation phase*). Every phase was fitted by a line fit, and the change in BRET signal was calculated as the distance of the last time point of the previous phase and the first point of its following phase. Hence, the agonist-induced change in BRET signal was calculated as the distance between the *baseline phase* and the *agonist phase*. The additional change induced by saturating concentrations of ACh was calculated as the distance between the *agonist phase* and the *saturation phase*. The maximum change in BRET signal was then calculated as the sum of the agonist-induced change and the additional change upon application of saturating concentrations of ACh. The change in BRET signal was normalized to the maximum change in BRET signal for every well.

## Radioligand displacement assay

The human $M_3R$ antagonist radioligand receptor binding assay was performed by Eurofins Cerep, Celle-Levescault, France (item 95). Competition between [$^3$H]4-DAMP (0.2 nM) and increasing concentrations of each compound for $M_3R$ in CHO whole cells was measured after incubation for 60 min at room temperature. Nonspecific binding was determined in the presence of 1 µM atropine, and 4-DAMP was used as a reference compound. Radioactivity was quantified by scintillation counting. Results are expressed as percent of control specific binding, that is, (specific binding/control-specific binding) × 100 obtained in the presence of the test compound.

## Statistics and data analysis

The radioligand displacement measurements performed by Eurofins are presented as mean ± SD of n experiments with two independent experiments. All other measurements are presented as mean ± SD of n experiments with at least three independent transfections. For statistical analysis, a paired Student's *t*-test as well as a one-way ANOVA with Tukey's multiple comparison test were conducted using GraphPad Prism 6.01. $EC_{50}$ values were obtained by using GraphPad Prism's nonlinear regression curve fit for the concentration-response curves, fitting the Hill equation with four parameters, variable slope, and by setting the bottom parameter to zero. For the radioligand displacement assay, $IC_{50}$ values were obtained from the inhibition curves and $K_i$ values were calculated using the Cheng and Prusoff method (*Cheng, 2001*).

For the evaluation of the offset kinetics, each cell was normalized to the maximum response of agonist, and a one-phase exponential decay for the treatment with ACh and a two-phase decay for the treatment together with the inhibitors were fitted using GraphPad Prism 8.3.0. The following equation was used for fitting the one-phase decay:

$Y = (Y0 - Plateau) \cdot e^{(-K \cdot x)} + Plateau$.

The following equation was used for fitting the two-phase decay:

$$SpanFast = (Y0 - Plateau) \cdot \%Fast \cdot .01$$

$$SpanSlow = (Y0 - Plateau) \cdot (100 - \%tFast) \cdot .01$$

$$Y = Plateau + SpanFast \cdot e^{(-KFast*x)} + SpanSlow \cdot e^{(-KSlow \cdot x)}$$

where Y0 is the y value when x is zero and *Plateau* is the y value at infinite times. For both conditions, the Y0 was constrained to 1 and the Plateau to 0.

The concentration-response data for $G_q$ activation, arrestin3 recruitment, and GRK2 recruitment to the $M_3R$ were analyzed using the operational model of agonism by **Black and Leff, 1997**. The following equation was used for fitting and calculation of $\tau$ :

$$Y = \frac{[A]^n \cdot \tau^n \cdot E_{max}}{[A]^n \cdot \tau^n + ([A] + K_i)^n}.$$

where $[A]$ is the molar concentration of agonist, $E_{max}$ is the theoretical maximum response of the system, $K_i$ is the equilibrium dissociation constant of the agonist-receptor complex, $n$ represents a slope factor, and $\tau$ is the operational factor of efficacy. The following equation was implemented in GraphPad Prism:

$$Y = \frac{(10^x)^n \cdot \tau^n \cdot E_{max}}{(10^x)^n \cdot \tau^n + (10^x + K_i)^n}$$

where $x$ is the log of the molar concentration of agonist $[A]$. Each $K_i$ value was set individually to the $K_i$ value determined by the competition binding experiment for each agonist. Because the response of GRK2 and arrestin3 recruitment for each agonist was normalized to the maximum response of ACh, $E_{max}$ was set to 1. For $G_q$ activation, where each agonist concentration was normalized to the maximum response of ACh in the same well, $E_{max}$ was set to the maximum response of ACh, to 98.63. One $n$ value was determined for all agonists, and $n$ and $\tau$ were constrained to be greater than 0. The fitted value of $\tau$ was used to calculate $log(\frac{\tau}{K_i})$ for each agonist. ACh was used as a reference compound that all other agonists were scaled to, thus allowing the calculation of $\Delta log(\frac{\tau}{K_i})$.

## Acknowledgements

We thank Sandra Engel for technical assistance and Daniel Hilger for critical comments on the manuscript. We thank the German Research Foundation DFG for Emmy Noether fellowship KO4095/1-1, Heisenberg professorship grants KO4095/4-1 and KO4095/5-1 (to PK) and grants for the research group FOR 2372, 290827466 (to GK)/290847012 (to EK).

## Additional information

### Funding

| Funder | Grant reference number | Author |
|---|---|---|
| Deutsche Forschungsgemeinschaft | KO-4095/1-1 | Peter Kolb |
| Deutsche Forschungsgemeinschaft | KO4095/4-1 | Peter Kolb |
| Deutsche Forschungsgemeinschaft | KO4095/5-1 | Peter Kolb |
| Deutsche Forschungsgemeinschaft | FOR 2372 290827466 | Gabriele König |
| Deutsche Forschungsgemeinschaft | FOR 2372 290847012 | Evi Kostenis |

The funders had no role in study design, data collection and interpretation, or the decision to submit the work for publication.

### Author contributions

Anja Flöser, Data curation, Formal analysis, Investigation, Methodology, Writing – original draft; Katharina Becker, Conceptualization, Data curation, Formal analysis, Investigation; Evi Kostenis, Funding acquisition, Resources, Writing – review and editing; Gabriele König, Funding acquisition, Resources; Cornelius Krasel, Methodology, Resources; Peter Kolb, Conceptualization, Funding acquisition,

Methodology, Project administration, Supervision, Writing – original draft, Writing – review and editing; Moritz Bünemann, Conceptualization, Methodology, Project administration, Resources, Supervision, Validation, Writing – original draft, Writing – review and editing

### Author ORCIDs
Anja Flöser  http://orcid.org/0000-0002-0876-1184
Katharina Becker  http://orcid.org/0000-0003-4104-7844
Peter Kolb  http://orcid.org/0000-0003-4089-614X
Moritz Bünemann  http://orcid.org/0000-0002-2259-4378

### Decision letter and Author response
Decision letter https://doi.org/10.7554/eLife.58442.sa1
Author response https://doi.org/10.7554/eLife.58442.sa2

---

## Additional files

### Supplementary files
• Transparent reporting form

### Data availability
All data generated or analysed during this study are included in the manuscript and supporting files.

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
