## [Editor Report]

This paper investigates the molecular mechanism of ligand bias in G protein-coupled receptors, specifically the M_3_ muscarinic receptor, which is the property that different receptor agonists favor activation of G protein signaling vs. arrestin-mediated signaling. The interaction with arrestin is promoted by receptor phosphorylation by G protein-coupled receptor kinases (GRK), and GRK recruitment is expected to influence arrestin-recruiting activity of a particular ligand. However, the possibility of a distinct agonist-dependent receptor conformation on GRK association has not been investigated. This study demonstrates that the G protein G_q_ and GRK2 appear to interact with a similar M_3_ receptor conformation, whereas arrestin3 interacts with a distinct conformation. This represents a significant advance in understanding the mechanism of ligand bias in G protein-coupled receptors.

---

## [Decision Letter]

**Decision letter after peer review:**

Thank you for submitting your article "Disentangling bias between G*_q_* activation, GRK2 recruitment, and arrestin3 recruitment at the M_3_R" for consideration by *eLife*. Your article has been reviewed by 3 peer reviewers, and the evaluation has been overseen by a Reviewing Editor and Richard Aldrich as the Senior Editor. The reviewers have opted to remain anonymous.

The reviewers have discussed the reviews with one another and the Reviewing Editor has drafted this decision to help you prepare a revised submission.

Summary:

This paper uses fluorescent probes attached to different the M3 muscarinic receptor (M3R) interacting proteins Gq, arrestin and GRK2 in cellular BRET assays to examine bias among different M3R ligands. Bias is typically discussed in terms of preferential activation of the partner G protein vs. arrestin. GRK recruitment/activity is important for arrestin binding and is expected to influence arrestin-recruiting activity of a particular ligand. However, the possibility of a distinct agonist-dependent receptor conformation on GRK association has not been investigated. This study attempts to answer this question.

Essential revisions:

While the work addresses a very important problem and is technically excellent, all of the reviewers agree that the data do not decisively establish that the receptor adopts a conformation distinct from that which binds arrestin or G protein. As you will see in points 1-3 below, the major concern is that the experiments have not completely disentangled GRK and G protein interactions. If you are able to obtain data that convincingly shows a GRK interaction without the influence of G proteins (which would require mutants or some other way of getting rid of/maximally activating G proteins), then the work would provide new and important information regarding bias between GRK- and arrestin-preferring conformations. The reviewers recognize that this may be very hard to do, but there are some suggestions along these lines as outlined below.

1. The main difficulty encountered by the authors relates to the fact, as the same group showed convincingly in [33], that agonist-dependent FRET between active M3 receptors and GRK2 depends almost entirely on the interactions of GRK2 with Gqα and betagamma subunits, which translocate GRK2 to the plasma membrane. The authors make the argument that under the conditions of their experiments Gq is maximally activated by all of the agonists they used, therefore any differences in GRK2 recruitment efficacy must be due to differences in the direct interaction between GRK2 and activated receptors. Setting aside possible differences in G protein activation due to the presence of inserted fluorescent tags and the different preparations this is a reasonable argument. However, there are a number of alternative approaches that could be used to validate the findings. Adding a membrane anchor to GRK2, overexpressing G betagamma subunits alone, blocking Gq activation with small molecules or gene deletion, maximally activating Gq with GTPgammaS, or a combination of these approaches, would greatly increase confidence that the result is fundamentally correct. Since the authors observe essentially no bias between Gq activation and GRK2 recruitment with the ligands they tested (Figure 7C), this could be because (as the authors conclude) the receptor conformations that activate Gq are the same as those that recruit GRK2, or alternatively that the GRK2 signals they measure still reflect primarily the efficiency of Gq activation. This is indeed a very important question because of the possibility that arrestin bias is actually a reflection of GRK bias, therefore the equivalency of G protein activation and GRK recruitment (the main novel conclusion of the paper) deserves very stringent testing.

2. GRK2 also binds to activated Galphaq subunits, which may have an effect on the kinetics of G protein activation and/or deactivation (along with the Gbg binding). This seems to be the case because the presence of overexpressed GRK2 left-shifts the G protein activation dose response curves in Figure 3.

3. Comparison of the data from Figure 3A with Figure 7B, which is used by the authors to argue that they don't need to worry about G protein-dependent effects on GRK2 recruitment seems invalid because GRKs likely have less affinity for phosphorylated receptors (Pulvermuller et al. 1993), as is the case in 7B, and they would compete poorly with arretin-bound GPCRs (endogenous or not). Regardless, also not sure if it is valid to say that there should be no coupling between G protein activation efficacy versus GRK recruitment because their EC50 curves are not similar. The affinity is what it is, and could still be dependent the amount of G protein subunits liberated. Issues like this is what makes interpreting this data very complicated.

4. How do the expression levels of tagged proteins compare to the endogenous levels? Do the M3R-YFP receptors used in the GRK2 experiments express at the same level as the unlabeled M3R receptors used in the G protein activation experiments? Activating the Gq BRET constructs with M3R-YFP might also be useful to address this point. Can the authors comment on how GRK2 overexpression increases the potency of Gq activation?

5. The authors have not tested or understood how the endogenous GRK and arrestin isoforms might be adding to or competing with the tagged proteins, raising the possibility that bias is instead for a specific GRK or arrestin 2 versus arrestin 3. The activated M3R is phosphorylated by several GRKs other than GRK2, such as GRK3 and GRK6 (J Luo et al. M3 muscarinic acetylcholine receptor-mediated signaling is regulated by distinct mechanisms. Mol. Pharmacol. 2008). This latter work was also carried out with HEK293 cells. The Luo et al. paper needs to be cited, and the results of this paper need to be discussed in the context of the new data. How does the Luo et al. study affect the conclusions here?

6. Is there anything known that different muscarinic agonists recruit different GRKs to the receptor, leading to different phosphorylation patterns? This could explain the observed discrepancies in rank order of agonist potencies (GRK2 vs. arrestin3 recruitment). No attempt is made to verify that the M3R is getting phosphorylated at GRK2 specific sites.

7. The potential implications of this study for drug development efforts should be discussed in more detail.

8. The times that people normally cite for full GRK phosphorylation of GPCRs is on the order of minutes unless one is talking about rhodopsin. The full activation of the "GRK" interaction shown after application of agonist suggests that this is really a GRK-G protein interaction, not a GRK-GPCR interaction. If so, how would this affect interpretation? It would have been good to see how a non-Gbg dependent GRK would respond in this assay.

[Editors' note: further revisions were suggested prior to acceptance, as described below.]

Thank you for submitting your article "Disentangling bias between G*_q_*, GRK2 and arrestin3 recruitment at the M_3_ muscarinic acetylcholine receptor" for consideration by *eLife*. Your article has been reviewed by 2 peer reviewers, and the evaluation has been overseen by a Reviewing Editor and Richard Aldrich as the Senior Editor. The following individuals involved in review of your submission have agreed to reveal their identity: John JG Tesmer (Reviewer #2); Nevin A Lambert (Reviewer #3).

Essential revisions:

The reviewers agree that the paper has been greatly strengthened by the new experimental data. They do agree, however, that the important message of the paper is weakened by inclusion of the "mixed signal" data in figures 2 and 4A,B (see Reviewer 3 comments below), which by their nature do not cleanly decouple signals from different sources. These data do support the separation of direct receptor vs. G protein-dependent signals and emphasize the importance of the decoupled assays in Fig. 4C,D , so we ask that the paper be revised with those data provided as supplemental rather than main figures. You may also want to consider commenting on the kinetics mentioned by reviewer 3.

*Reviewer #2:*

I am impressed with the robust experimental response to the two reviews and I think it is an example where things did dramatically improve in terms of rigor. I am satisfied with their response to my comments. I think this is a very interesting comparative study and I cannot think of many better ways to disentangle how bias manifests itself at the M3R. One could quibble onwards because of the complexities of the system and the fact that overexpressed proteins were used, but I think this data should be seen by the community at large now.

*Reviewer #3:*

The authors have done a very nice job in addressing a most difficult problem, i.e. separating G protein-dependent and -independent GRK2 signals. They did this with a CAAX anchor and PTX/FR900359. I still suggest that Figures 1 and 2 are not necessary and do not add to the story, and in fact confuse the situation by showing "M3R-GRK2 interaction" that is clearly a mixed response that includes bystander FRET- the signal is largely free Gbetagamma-GRK, not receptor-GRK. Figure 4A and 4B are similarly confusing and unnecessary- why show FRET signals that are not interpretable?

The results in 4C and 4D are key, and should take center stage. The authors should also point out that the kinetics of the isolated G protein-independent GRK signals are fast, as would be expected for what is essentially a reporter of receptor conformation. In contrast, the mixed GRK signals in 4A, 4B and Figure 2 have slow decays, presumably because this is rate-limited by G protein deactivation (GTP hydrolysis). This kinetic difference supports their claim that they've successfully isolated the receptor-GRK interaction from G proteins. Along these same lines, it's worth noting that the presumed bystander signal after PTX/FR900358 in Figure 4S1 C is also fast; the slower components due to G proteins get blocked by the PTX/FR. Indeed, the decay of the untreated trace in this figure could be fitted to two exponentials, the slower of the two would be consistent with GTP hydrolysis, the faster with receptor deactivation. A brief kinetic analysis of this sort would strengthen the paper.

---

## [Author Response]

Essential revisions:While the work addresses a very important problem and is technically excellent, all of the reviewers agree that the data do not decisively establish that the receptor adopts a conformation distinct from that which binds arrestin or G protein. As you will see in points 1-3 below, the major concern is that the experiments have not completely disentangled GRK and G protein interactions. If you are able to obtain data that convincingly shows a GRK interaction without the influence of G proteins (which would require mutants or some other way of getting rid of/maximally activating G proteins), then the work would provide new and important information regarding bias between GRK- and arrestin-preferring conformations. The reviewers recognize that this may be very hard to do, but there are some suggestions along these lines as outlined below.1. The main difficulty encountered by the authors relates to the fact, as the same group showed convincingly in [33], that agonist-dependent FRET between active M3 receptors and GRK2 depends almost entirely on the interactions of GRK2 with Gq α and betagamma subunits, which translocate GRK2 to the plasma membrane. The authors make the argument that under the conditions of their experiments Gq is maximally activated by all of the agonists they used, therefore any differences in GRK2 recruitment efficacy must be due to differences in the direct interaction between GRK2 and activated receptors. Setting aside possible differences in G protein activation due to the presence of inserted fluorescent tags and the different preparations this is a reasonable argument. However, there are a number of alternative approaches that could be used to validate the findings. Adding a membrane anchor to GRK2, overexpressing G betagamma subunits alone, blocking Gq activation with small molecules or gene deletion, maximally activating Gq with GTPgammaS, or a combination of these approaches, would greatly increase confidence that the result is fundamentally correct. Since the authors observe essentially no bias between Gq activation and GRK2 recruitment with the ligands they tested (Figure 7C), this could be because (as the authors conclude) the receptor conformations that activate Gq are the same as those that recruit GRK2, or alternatively that the GRK2 signals they measure still reflect the primarily the efficiency of Gq activation. This is indeed a very important question because of the possibility that arrestin bias is actually a reflection of GRK bias, therefore the equivalency of G protein activation and GRK recruitment (the main novel conclusion of the paper) deserves very stringent testing.

We thank the reviewer to bring up this important point. We spent a lot of time to clearly address the receptor-GRK2 interaction without the influence of activated G proteins. Since the experiments of M3R-GRK2 interaction by fully activated G proteins in presence of GTPyS, as suggested by the reviewers, did not show reliable alterations in FRET signals, we finally combined numerous approaches. To bypass the G protein dependent translocation (Wolters et al) of GRK2 to the plasma membrane, we introduced a CAAX-box at the C-terminus of the GRK2. Furthermore, we mutated the GRK2 at position D110 to an alanine in order to attenuate Gaq binding (Wolters et al). In addition, we performed the experiments without overexpression of Gaq and preincubated the cells with the Gi/o inhibitor PTX and the Gq inhibitor FR900365 to inhibit endogenous G proteins that could potentially influence GRK2 (Figure 4C,D, supplemental Figure 1A,B,C.). Thus, we removed all possibilities of GRK2-G protein interaction by a combination approach, as indicated by the reviewer. Importantly, even though the agonist-evoked FRET responses under complete Gi and Gq inhibition were markedly reduced, we were still able to resolve agonist-specific interactions between (D110A)GRK2-CAXX and M3R shown in Figure 4 C, D. Therefore, we have specifically measured GRK2 interaction with M3R without any influence of G proteins. Nevertheless, the rank order of agonist efficacy for the GRK2-M3R interaction is the same as for G protein interactions, as shown in Figure 4D.

2. GRK2 also binds to activated Galphaq subunits, which may have an effect on the kinetics of G protein activation and/or deactivation (along with the Gbg binding). This seems to be the case because the presence of overexpressed GRK2 left-shifts the G protein activation dose response curves in Figure 3.

We agree with the reviewer that GRK2 also binds to activated Gaq, which was already shown in [33]. After our control measurements regarding point 4 (different expression levels of wildtype receptors and YFP-tagged receptors), we decided to measure the direct Gq binding to the M_3_ receptor, determined by single cell FRET assays, instead of G protein activation (Figure 3). The G protein binding assay allows us to much more directly measure agonist efficacy. Furthermore, as described in response to point 1, we have now measured GRK2 interaction with the M3R in the absence of G protein activation, making it unnecessary to tease apart the exact reason of the left shift of the concentration response curves for of Gq protein activation by GRK2.

3. Comparison of the data from Figure 3A with Figure 7B, which is used by the authors to argue that they don't need to worry about G protein-dependent effects on GRK2 recruitment seems invalid because GRKs likely have less affinity for phosphorylated receptors (Pulvermuller et al. 1993), as is the case in 7B, and they would compete poorly with arretin-bound GPCRs (endogenous or not). Regardless, also not sure if it is valid to say that there should be no coupling between G protein activation efficacy versus GRK recruitment because their EC50 curves are not similar. The affinity is what it is, and could still be dependent the amount of G protein subunits liberated. Issues like this is what makes interpreting this data very complicated.

We totally agree with the reviewer that in the previous version of the manuscript we were not able to completely disentangle G protein activation and GRK2 recruitment, which was also shown in the measurement of different expression levels of wildtype receptor and YFP-tagged receptor (point 4), as suggested by the reviewers. However, in the revised version of our manuscript we have solved this issue by performing GRK2 recruitment assays under conditions of complete G protein inhibition (Figure 4 C, D) (as described above in point 1) to exclude the influence of activated G proteins on the GRK2 recruitment. In addition, we now added the M3R Gq-protein interaction assay as a direct measurement of agonist efficacy, which allows for more consistent comparison of efficacies. These major revisions allow for a much better disentanglement of the influence of G proteins on GRK2 and vice versa. We are not speculating about possible differential dynamics of GRK2 binding to phosphorylated and unphosphorylated receptors. However, as published previously, there is no major difference of amplitude and kinetics of GRK2 binding and dissociation to M3R, whether a dominant negative GRK2 (K220R) is used or the GRK2-wt (Wolters et al., 2015).

4. How do the expression levels of tagged proteins compare to the endogenous levels? Do the M3R-YFP receptors used in the GRK2 experiments express at the same level as the unlabeled M3R receptors used in the G protein activation experiments? Activating the Gq BRET constructs with M3R-YFP might also be useful to address this point. Can the authors comment on how GRK2 overexpression increases the potency of Gq activation?

We are glad to follow the reviewers’ suggestion to address this important point. We performed BRET-based measurements of Gq activation for M3 wt and M3-YFP (Author response image 1). Due to the right shift in the concentration response curves of M3-YFP relative to M3 wt, we can no longer claim that G proteins have been fully activated in our GRK2 recruitment measurements. Therefore, we performed, as described above (point 1 and 2), a G protein-independent GRK2 recruitment assay and the direct G protein binding to the receptor to directly compare agonist efficacies (Figure 3 vs Figure 4 C-D), which is now the primary data set for comparing bias. That being said, the main conclusions about the different behavior of the different ligands stand also with this data set.

**Author response image 1. sa2fig1:** Gq activation of M3 wt and M3-YFP was measured with a BRET-based assay in HEK293T cells. (A) Concentration-response curves of Gq activation with GRK2 overexpression were plotted for M_3_R wt and M_3_R-YFP. (B) M_3_R wt and (C) M_3_R-YFP Gq activation were measured with distinct concentrations of every agonist as indicated. After stimulating every well with agonist, a saturating concentration of ACh (50 µM) was applied. The BRET ratio after agonist application was normalized to the maximum response of ACh (50 µM) of every well. All data are plotted as mean values ± SD for each condition.

As the rank order of agonist efficacies to activate Gq-proteins is the same for the tagged versus untagged receptor (Author response image 1), and the G protein activation data all moved to the supplemental figures we decided to show this dataset only to the reviewers. If requested, we could show this dataset also in the supplemental figures.

Regarding the reviewers’ question “Can the authors comment on how GRK2 overexpression increases the potency of Gq activation?” we admit that this is a bit complicated: As described by Wolters et al., the FRET decrease is markedly enhanced due to simultaneous binding of Gaq and Gbg to GRK2 and in addition the recovery of the FRET between Gaq and Gbg after agonist withdrawal is a bit slowed. This could lead to an apparent left shift of the concentration-response curve for Gq activation. However, this issue is of much less importance for our conclusions due to the implementation of the direct measurement of Gq interaction with M3R. Therefore, we prefer to not discuss this issue in the manuscript.

5. The authors have not tested or understood how the endogenous GRK and arrestin isoforms might be adding to or competing with the tagged proteins, raising the possibility that bias is instead for a specific GRK or arrestin 2 versus arrestin 3. The activated M3R is phosphorylated by several GRKs other than GRK2, such as GRK3 and GRK6 (J Luo et al. M3 muscarinic acetylcholine receptor-mediated signaling is regulated by distinct mechanisms. Mol. Pharmacol. 2008). This latter work was also carried out with HEK293 cells. The Luo et al. paper needs to be cited, and the results of this paper need to be discussed in the context of the new data. How does the Luo et al. study affect the conclusions here?

We agree with the reviewer that we have not tested how the endogenous GRK and arrestin isoforms add or compete with the tagged proteins, but the influence on our measurements of endogenous isoforms should be negligible through the overexpression of our proteins. Moreover, Hosey et al. (1995) showed that GRK2 and GRK3 were the most effective kinases at the M3 receptor. Additionally, Luo et al. (2008) suggested that it is primarily the GRK2 that regulates the M3 receptor through its ability to interact with Gq. With our data we can assume that the GRK2-M3 receptor interaction is essentially based on the regulation via G proteins. The difference between agonist efficacies of wildtype GRK2-M3 receptor interaction (Figure 4 B) and G protein independent GRK2-M3 receptor interaction (Figure 4 D) show the impact of G proteins on the translocation of the GRK2 to the plasma membrane. It was also shown in Luo et al. that arrestin-3 appears to play a larger role in terminating signaling in response to agonist exposure at the M3 receptor compared to arrestin-2. (We included a sentence indicating this fact: “We used in our study GRK2 and arrestin-3 isoforms, which appear to be the most regulating isoforms at the M3 receptor as shown in Luo et al.”). We discuss this issue and the respective limitations in the Discussion section.

6. Is there anything known that different muscarinic agonists recruit different GRKs to the receptor, leading to different phosphorylation patterns? This could explain the observed discrepancies in rank order of agonist potencies (GRK2 vs. arrestin3 recruitment). No attempt is made to verify that the M3R is getting phosphorylated at GRK2 specific sites.

We thank the reviewer for bringing up this point. It was shown in Butcher et al. (2011) that different agonists can induce phosphorylation of the M3 receptor preferentially at specific sites. However, in our study we avoided complications by agonist-specific phosphorylation patterns or extent by always pre-exposing the cells to the full agonist acetylcholine, just before we applied one of the specific investigated agonists. Thus, our experimental protocol with single cell FRET recording, which allows for a fast solution exchange, allowed to specifically compare agonist-specific recruitment of arrestin 3 to M3R without interference with effects derived from agonist-specific G protein activation or GRK recruitment (Figure 4 A, C). Thus, even without actually measuring bulk receptor phosphorylation (it is in our opinion impossible to mirror our experimental conditions in a phosphorylation assay), we can conclude that the difference in agonist efficacies is not based on different phosphorylation patterns of the M3 receptor.

7. The potential implications of this study for drug development efforts should be discussed in more detail.

We thank the reviewers for their helpful suggestion. Therefore, we stated in the last part of the discussion: "The finding that different receptor ligands can differentially affect binding of G proteins, arrestins and GRKs opens up the potential for future drug development to specifically direct signaling in one or the other direction."

8. The times that people normally cite for full GRK phosphorylation of GPCRs is on the order of minutes unless one is talking about rhodopsin. The full activation of the "GRK" interaction shown after application of agonist suggests that this is really a GRK-G protein interaction, not a GRK-GPCR interaction. If so, how would this affect interpretation? It would have been good to see how a non-Gbg dependent GRK would respond in this assay.

We agree with the reviewer that our GRK2 recruitment measurements were still affected by activated G proteins. As explained above the new measurements of the G protein independent GRK2–M3 interaction showed reduced overall amplitudes and reduced agonist-induced responses normalized to the one of acetylcholine, indicating the impact of G proteins on GRK2 recruitment (Figure 4 A-D). However, having eliminated any influence from the G protein, we still find the same rank order of agonist efficacies to recruit GRK2 to the receptor.

[Editors' note: further revisions were suggested prior to acceptance, as described below.]

Reviewer #3:The authors have done a very nice job in addressing a most difficult problem, i.e. separating G protein-dependent and -independent GRK2 signals. They did this with a CAAX anchor and PTX/FR900359. I still suggest that Figures 1 and 2 are not necessary and do not add to the story, and in fact confuse the situation by showing "M3R-GRK2 interaction" that is clearly a mixed response that includes bystander FRET- the signal is largely free Gbetagamma-GRK, not receptor-GRK. Figure 4A and 4B are similarly confusing and unnecessary- why show FRET signals that are not interpretable?

We agree with the reviewer that Figures 1 and 2 are not really necessary and do not add to the story. Therefore, we moved both figures to the supplement, resulting in Figure 1—figure supplement 1 and Figure 1—figure supplement 3. Additionally, we fully agree that it is confusing to show the M3R-GRK2 interaction in Figure 2 and in Figure 4A, B. Thus, we exchanged the example measurement of M3R-GRK2 interaction to M3R-(D110A)GRK2-CAAX interaction shown in the new Figure 1—figure supplement 3B. Figure 4A and B were moved to Figure 2—figure supplement 1 A and B.

The results in 4C and 4D are key, and should take center stage. The authors should also point out that the kinetics of the isolated G protein-independent GRK signals are fast, as would be expected for what is essentially a reporter of receptor conformation. In contrast, the mixed GRK signals in 4A, 4B and Figure 2 have slow decays, presumably because this is rate-limited by G protein deactivation (GTP hydrolysis). This kinetic difference supports their claim that they've successfully isolated the receptor-GRK interaction from G proteins. Along these same lines, it's worth noting that the presumed bystander signal after PTX/FR900358 in Figure 4S1 C is also fast; the slower components due to G proteins get blocked by the PTX/FR. Indeed, the decay of the untreated trace in this figure could be fitted to two exponentials, the slower of the two would be consistent with GTP hydrolysis, the faster with receptor deactivation. A brief kinetic analysis of this sort would strengthen the paper.

We are glad to follow the reviewer’s suggestion to point out the differences in offset-kinetics in presence and inhibition of G proteins. We added a half sentence to the results part stating now on page 4 : “… Under conditions of complete Gq and Gi inhibition (see for control Figure 2—figure supplement 1D,E), amplitudes of agonist-evoked FRET increases were reduce and offset kinetics after agonist withdrawal were substantially faster (compare Figure 2A and Figure 2 supplement 1A). …..” Furthermore, as suggested we analysed the offset-kinetics of the Gβγ/GRK2 interaction with and without the inhibitors (Figure 2—figure supplement 1 E, F and G) and stated in the results: “Kinetic analysis of agonist-evoked FRET between Gβγ and GRK2 with or without pre-treatment with PTX and FR900359 showed fast one phased decays for signals measured with pre-treatment and slower two phased decays for signals measured without pre-treatment (Figure 2—figure supplement 1 F, G).” Finally, we completed the sentence in the discussion: “However, the overall FRET amplitudes and also the agonist responses normalized to the one of ACh were considerably larger and their overall kinetics considerably slower in the absence of G protein inhibition, demonstrating the contribution of G proteins to translocating GRK2 to the plasma membrane.”

Again, we would like to thank the reviewers for their many helpful comments and suggestions, which have indeed substantially improved the manuscript and hope that the revised manuscript can be accepted for publication in *eLife*.